# *BRCA1/2* Serves as a Biomarker for Poor Prognosis in Breast Carcinoma

**DOI:** 10.3390/ijms23073754

**Published:** 2022-03-29

**Authors:** Tong Yi Jin, Kyoung Sik Park, Sang Eun Nam, Young Bum Yoo, Won Seo Park, Ik Jin Yun

**Affiliations:** 1Department of Surgery, Konkuk University School of Medicine, Seoul 05030, Korea; jintongyi@konkuk.ac.kr (T.Y.J.); 20090055@kuh.ac.kr (S.E.N.); 0117652771@kuh.ac.kr (Y.B.Y.); ijyun@kuh.ac.kr (I.J.Y.); 2Research Institute of Medical Science, Konkuk University School of Medicine, Seoul 05030, Korea; 3Department of Surgery, Konkuk University Medical Center, Seoul 05030, Korea; 4Department of Surgery, Kyung Hee University School of Medicine, Seoul 02447, Korea; pwsmd@hanmail.net

**Keywords:** *BRCA1*/*2*, breast cancer, multiomics, prognostic biomarkers

## Abstract

*BRCA1/2* are breast cancer susceptibility genes that are involved in DNA repair and transcriptional control. They are dysregulated in breast cancer, making them attractive therapeutic targets. Here, we performed a systematic multiomics analysis to expound *BRCA1*/*2* functions as prognostic biomarkers in breast cancer. First, using different web-based bioinformatics platforms (Oncomine, TIMER 2.0, UALCAN, and cBioportal), the expression of *BRCA1/2* was assessed. Then, the R package was used to analyze the diagnostic value of *BRCA1/2* in patients. Next, we determined the relationship between *BRCA1/2* mRNA expression and prognosis in patients (PrognoScan Database, R2: Kaplan Meier Scanner and Kaplan–Meier Plotter). Subsequently, the association of *BRCA1/2* with mutation frequency alteration and copy number alterations in breast cancer was investigated using the cBioportal platform. After that, we identified known and predicted structural genes and proteins essential for *BRCA1/2* functions using GeneMania and STRING db. Finally, GO and KEGG pathway enrichment analyses were performed to elucidate the potential biological functions of the co-expression genes of *BRCA1/2*. The *BRCA1/2* mRNA level in breast cancer tissues was considerably higher than in normal tissues, with AUCs of 0.766 and 0.829, respectively. Overexpression of *BRCA1/2* was significantly related to the worse overall survival (*p* < 0.001) and was correlated to clinicopathological characteristics including lymph nodes, estrogen receptors, and progesterone receptors (*p* < 0.01). The alteration frequencies of both the gens have been checked, and the results show that *BRCA1* and *BRCA2* show different alteration frequencies. Their mutation sites differ from each other. GO and KEGG showed that *BRCA1/2* was mainly enriched in catalytic activity, acting on DNA, chromosomal region, organelle fission, cell cycle, etc. The 20 most frequently changed genes were closely related to *BRCA1/2*, including *PALB2* and *RAD51* relatively. Our study provides suggestive evidence of the prognostic role of *BRCA1/2* in breast cancer and the therapeutic target for breast cancer. Furthermore, *BRCA1/2* may influence BRCA prognosis through catalytic activity, acting on DNA, chromosomal regions, organelle fission, and the cell cycle. Nevertheless, further validation is warranted.

## 1. Introduction

Breast cancer affects a staggering number of 1.7 million women worldwide each year, which is the main leading cause of cancer death in women [1,2]. Five to ten percent of breast cancers are considered to be related to genetic predispositions, and some breast cancers are caused by genetic dominant susceptibility genes, which have a high risk of disease [3].

*BRCA1/2* are tumor suppressors that are essential for the faithful repair of double-strand DNA breaks by homologous recombination. However, *BRCA1/2* mutation occurs frequently in breast cancer [4]. The tumor suppressor genes *BRCA1/2*, which are found on chromosomes 17q and 13q, respectively, and encode factors that restrict cell development, were identified in the early 1990s [5,6,7,8,9]. *BRCA1* encodes a nuclear phosphoprotein, which acts as a tumor suppressor gene through maintaining genomic stability [3,10]. *BRCA1* is also involved in other cellular functions that are important in maintaining genomic integrity, including mitotic spindle assembly, centrosome replication, cell cycle control, and chromatin remodeling at dsDNA break sites [4]. *BRCA2* is a 27-exon gene involved in genome stability, particularly in the homologous recombination (HR) pathway, which repairs double-stranded DNA breaks [3]. *BRCA2*- related cancers, unlike *BRCA1*, frequently express estrogen and progesterone receptors and have many of the same features as sporadic breast cancer [11]. Women with germline *BRCA1* or *BRCA2* (*BRCA1/2*) mutations have a markedly increased risk of breast and ovarian cancer compared with the general population [12,13,14]. *PALB2* is a protein-coding gene that is associated with both *BRCA1* and *BRCA2*; it plays a critical role in homologous recombination repair (HRR) through its ability to recruit *BRCA2* and *RAD51* to DNA breaks [15]. *BRCA1* promotes the concentration of *PALB2* and *BRCA2* at DNA-damage sites, and the interaction between *BRCA1* and *PALB2* is important for homologous recombination repair [15].

In recent years, *BRCA1/2* has attracted considerable attention as a biomarker for predicting cancer prognosis. The identification of biomarkers, such as *BRCA1/2*, *PIK3CA* and *RSK2*, plays an important role in promoting the precise treatment of TNBC [16]. In this study, we assessed the significance of *BRCA1/2* expression in breast cancer using multiple online databases and annotation tools. By accessing and analyzing all existing gene expression data, the expression patterns, functions, and prognostic values of *BRCA1/2* in breast cancer were comprehensively investigated. These results will facilitate the understanding of the prognostic value of *BRCA1/2* in breast cancer in humans.

## 2. Results

### 2.1. BRCA1/2 Expression in Human Breast Cancer

The Oncomine database was used to analyze the mRNA levels of *BRCA1/2* between tumor tissues and normal tissues. The results showed that *BRCA1/2* were overexpressed in breast cancer as compared to their expression levels in normal tissues (Figure 1a). To further evaluate the differential expression of *BRCA1/2*, we compared its expression levels in the TCGA dataset using the TIMER2.0 database. As shown in Figure 1b, expression of *BRCA1/2* in breast cancer tissues is significantly higher than in normal tissue (Figure 1b). *BRCA1* was overexpressed in intraductal cribriform breast adenocarcinoma patients in the TCGA database, with a fold change of 3.454 and a *p* value of 1.92 × 10^−6^. *BRCA2* was also overexpressed in mucinous breast cancer, with a fold change of 3.750 and a *p* value of 3.22 × 10^−5^. The mRNA levels of *BRCA2* were significantly increased in patients with invasive ductal breast carcinoma and invasive lobular breast carcinoma, with fold changes of 3.214 and 2.124, and *p* values of 1.46 × 10^−25^ and 9.36 × 10^−9^, respectively (Table 1). Furthermore, it was discovered that the expression of *BRCA1/2* mRNA in breast cancer tissues was significantly higher than that in normal tissues (Figure 2a), and the change rates of *BRCA1/2* mRNA in BRCA were 9% and 6%, respectively (Figure 2b), and the AUC of *BRCA1/2* was 0.766 and 0.830, respectively (Figure 3), using UALCAN, cBioPortal, and the TCGA database.

### 2.2. Clinicopathological Parameters of BRCA1/2 in BRCA Patients

In the TCGA database, overexpression of *BRCA1/2* was correlated with clinicopathological features and prognosis of BRCA patients (Table 2). Overexpression of *BRCA1* was related to lymph node stage, ER status, PR status and PAM 50 subtypes (*p* < 0.05). Overexpression of *BRCA2* was related to age, lymph node stage, ER stage, PR status, HER-2 status, and PAM 50 subtypes (*p* < 0.01) (Table 2).

### 2.3. Prognostic Value of BRCA1/2 mRNA Expression in BRCA Patients

The prognostic value of *BRCA1/2* expression in breast cancer was investigated using the PrognoScan, Kaplan–Meier Plotter and R2 databases to explain how *BRCA1/2* affects the prognostic characteristics of breast cancer patients and the associations between variations in the expression of *BRCA1/2* genes and clinical outcomes. We observed a positive correlation between *BRCA1/2* overexpression and poor patient survival in breast cancer (Figure 4, *p* < 0.01). Kaplan–Meier Plotter’s survival analysis showed that the overall survival of breast cancer patients was poor in both *BRCA1/2* overexpression. In PrognoScan and R2-Kaplan Meier Scanner, overexpression of *BRCA1/2* was associated with poor prognosis.

### 2.4. The Relationship of BRCA1/2 Mutations and Copy Number Alterations (CNAs) in Breast Cancer

We individually examined the genetic alterations of *BRCA1/2* in breast cancer using cBioPortal. Based on the Breast Invasive Carcinoma (TCGA, Cell 2015) samples and studies, with clinical data from 816 patients, we queried the database for *BRCA1/2*. The results showed that there was a significant difference in *BRCA1/2* alteration frequency in breast cancer. As presented in Figure 5, the alteration frequency of *BRCA1/2* was determined. It included mutations (green), amplification (red), deep deletions (blue), and multiple alterations (grey) (Figure 5a). Of the queried samples of *BRCA1*, 76 (9%) samples were associated with an altered gene set or pathway, resulting in a somatic mutation frequency of 2.2%. There were 18 mutation sites detected located from amino acids 0 to 1863 of the *BRCA1* protein, including 12 missenses, 4 truncating, and 1 inframe mutation. Of the queried samples of *BRCA2*, 76 (9%) samples were associated with an altered gene set or pathway, resulting in a somatic mutation frequency of 2.1%. In *BRCA2*, a total of 19 mutation sites were detected between amino acids 0 to 3418 of the *BRCA2* protein, including 10 missenses, 7 truncating, and 1 inframe mutation (Figure 5b).

### 2.5. Co-Expression Analysis of BRCA1/2

There were 3919 positively correlated genes and 37 negatively correlated genes for *BRCA1* in the TCGA transcriptome database, and 3720 positively correlated genes and 340 negatively correlated genes for *BRCA2*. The top 10 co-expressed genes with positive and negative correlations to *BRCA1/2* are shown in the form of heat maps (Figure 6a,b). In addition, the Venn diagram indicates that there were 958 intersections of BRCA1/2 co-expressing genes (Figure 6c).

### 2.6. Analysis of GO and KEGG Pathway for BRCA1/2

To further understand the potential role of *BRCA1/2* in BRCA, GO and KEGG analyses were performed on *BRCA1/2* co-expressed genes. The results revealed that *BRCA1/2* co-expressed genes were mainly involved in catalytic activity, acting on DNA, chromosomal regions, and organelle fission (Figure 7). KEGG pathway analysis displayed that *BRCA1/2* co-expressed genes were enriched in regulating the cell cycle, RNA transport, and the Fanconi anemia pathway (Figure 8).

### 2.7. Identification of BRCA1/2 Interacting Genes and Proteins and Genetic Alterations

We constructed the gene–gene interaction networks for *BRCA1/2* and the altered neighboring genes by using GeneMania. The results showed that 20 of the most frequently altered genes were closely correlated with *BRCA1*, including *PALB2*, *RAD51*, and *PARP1* (Figure 9a). There were 20 most frequently altered genes that were closely correlated with *BRCA2*, including *PALB2*, *RAD51*, and *SEM1* (Figure 9b). The A protein–protein interaction (PPI) networks of *BRCA1/2* were generated using the STRING database. The result of the PPI network of *BRCA1* showed that there were 45 edges and 11 nodes, including PALB2, RAD51, and TP53 (Figure 9c). There were 50 edges and 11 nodes in the result of the PPI network of *BRCA2*, including *RAD51*, *BARD1* and *XRCC3* (Figure 9d).

## 3. Discussion

The prognostic importance of *BRCA1/2* in breast cancer was investigated in this research. *BRCA1/2* are high-risk breast cancer susceptibility genes with 11–12-fold elevated relative risks in the general population for women [17]. A previous study showed that the probability of breast cancer in *BRCA1* mutation carriers is 57–65%, while the probability of breast cancer in *BRCA2* mutation carriers is 45–49% [4]. In the past, many studies have explored the relationship between *BRCA1/2* mutation and survival. However, due to the relatively low mutation rate, detection rate, and survival bias of *BRCA1/2* mutation carriers, the evidence of low survival rate of *BRCA1/2* mutation carriers is still inconclusive [18]. With the aim of comprehensively covering the topic of the prognostic role of *BRCA1/2* mutations in breast cancer, in this study, we first analyzed the expression of *BRCA1/2* in breast cancer tissues and normal tissues. The results showed that *BRCA1/2* were overexpressed in breast cancer tissues than in normal tissues, and we obtained the relationship of *BRCA1/2* expression, mutation, and survival data in breast cancer from TCGA data analysis, which further verified the above results.

A previous study has reported that *BRCA1* and *BRCA2* carriers have varied clinical characteristics [19]. Herein, our study found that overexpression of *BRCA1* was related to lymph node stage, ER status, PR status and PAM 50 subtypes (*p* < 0.05), and overexpression of *BRCA2* was related to age, lymph node stage, ER stage, PR status, HER-2 status, and PAM 50 subtypes (*p* < 0.01).

The prognostic role of *BRCA1/2* mutational status in breast cancer patients is unclear. According to previous research, BRCA1/2 mutation carriers had a higher risk of mortality from breast cancer (HR 1.44, 95% CI: 1.05–1.97) and distant metastases (HR 1.82, 95% CI: 1.05–3.16) than sporadic/BRCA-negative individuals [20]. The estimated average cumulative risk of breast cancer to age 70 years was estimated to be 52% (95% CI, 26–69%) for *BRCA1* mutation carriers and 47% (95% CI, 29–60%) for *BRCA2* mutation carriers [21]. Nevertheless, whether *BRCA1/2* mutations are associated with poor prognosis in breast cancer remains controversial. Some studies suggest that breast cancer *BRCA1/2* mutation carriers have poorer overall survival (OS) [5,22,23,24,25,26,27,28,29,30]. Other studies have shown that *BRCA1/2* mutation carriers have better survival than non-carriers [5,31,32,33]. Several studies have showed that *BRCA1/2* mutations occur at scattered sites and occur as missense mutations, of which will especially influence those situated in exon-encoding domains that interact with *BRCA1*-binding proteins, such as *BARD1*, *BRIP1* and *PALB2*, which (along with *RAD51C*, *RAD51D* and possibly RAP80 and FAM175A, encoding Abraxas) are also breast and/or ovarian cancer-susceptibility genes [34,35].

In this study, we found that overexpression of *BRCA1*/*2* was significantly related to worse overall survival (*p* < 0.001). As a result, these genes might be used as novel anticancer targets. GO analysis showed that *BRCA1/2* co-expressed genes were mainly involved in catalytic activity, acting on DNA, chromosomal regions, and organelle fission. KEGG pathway analysis displayed that *BRCA1/2* co-expressed genes were enriched in regulating the cell cycle, RNA transport, and the Fanconi anemia pathway. In gene–gene interaction networks, 20 of the most frequently altered genes were closely correlated with *BRCA1*, including *PALB2*, *RAD51*, and *PARP1*. There were 20 most frequently altered genes that were closely correlated with *BRCA2*, including *PALB2*, *RAD51*, and *SEM1*. In the PPI network, there were 45 edges and 11 nodes in *BRCA1*, including *PALB2*, *RAD51*, and *TP53*. There were also 50 edges and 11 nodes in *BRCA2*, including *RAD51*, *BARD1* and *XRCC3*. *PALB2* was initially identified as a protein that interacts with *BRCA2* and plays a crucial role in the *BRCA2* genome caretaker function and was subsequently shown to also interact with *BRCA1* [15,36,37,38]. Biallelic germline loss-of-function mutations in *PALB2* cause Fanconi’s anemia, whereas monoallelic loss-of-function mutations are associated with an increased risk of breast cancer [39]. *RAD51* is a strand transferase that helps DNA strands exchange with undamaged homologous chromatids by polymerizing nucleoprotein filaments on single-stranded DNA [40,41]. Overexpression of *RAD51* is associated with tumor aggressiveness and is known to confer treatment resistance in a variety of tumors, including breast cancer [42]. *PARP1* has been shown to interact with DNA polymerase α and multiprotein DNA replication complexes [43,44,45]. *BRCA1* and *BARD1* both bind DNA and interact with *RAD51*, and *BRCA1-BARD1* enhances the recombinase activity of *RAD51* [35].

The present study used a systematic multiomics approach to investigate *BRCA1/2* functions as prognostic biomarkers in breast cancer. Although we investigated the correlation between *BRCA1/2* and breast cancer patients, some limitations remain. Nevertheless, there is a lack of in-depth insight into the relevant mechanisms involved, and further research is required. Overall, our study provides suggestive evidence of the prognostic role of *BRCA1/2* in breast cancer and the therapeutic target for breast cancer. Furthermore, *BRCA1/2* may influence BRCA prognosis through catalytic activity, acting on DNA, chromosomal regions, organelle fission, and the cell cycle. Nevertheless, further validation is warranted.

## 4. Material and Methods

### 4.1. Patient Datasets

The Cancer Genome Atlas (TCGA) database (https://www.cancer.gov/about-nci/organization/ccg/research/structural-genomics/tcga, accessed on 10 February 2022) is a publicly funded project initiated by the United States government with the goal of classifying and discovering the major genomic alterations that cause cancer to create a comprehensive The Cancer Genome Atlas database that includes more than 20,000 molecular characterizations of primary cancers and matched normal samples for 33 cancer types [46]. It was used to download all the raw data for *BRCA1/2*, including transcriptome RNA-seq data and clinical information.

### 4.2. Transcript Expression Analysis Using Oncomine Platform

The Oncomine database (https://www.oncomine.org/resource/login.html, accessed on 10 February 2022) is a web-based data mining platform that includes 264 separate datasets and aims to collect, standardize, evaluate, and distribute transcriptomic cancer data for biomedical research [47]. The expression levels of *BRCA1*/*2* in breast cancer were retrieved from the Oncomine platform. The *p* value 1 × 10^−4^, fold-change 2, and gene ranking in the top 10% were used to calculate the fold change in the expression of *BRCA1/2* in clinical cancer specimens compared to normal controls. The co-expression profile of the *BRCA1*/*2* in breast cancer was shown as a heat map.

### 4.3. TIMER 2.0 Analysis

Tumor IMmune Estimation Resource 2.0 (http://timer.cistrome.org, accessed on 10 February 2022) is a comprehensive resource for systematical analysis of immune infiltrates across diverse cancer types [48]. In this study, TIMER2.0 was used to determine the expression of *BRCA1*/*2* in breast cancer and normal tissues across all TCGA cohorts.

### 4.4. Expression of BRCA1/2 in BRCA Based on Sample Types by UALCAN

Data on TCGA have been used for *BRCA1*/*2* expression analysis. There are 1211 patients enrolled in this data profile, including 114 of normal and 1097 of tumors in patients with breast cancer. The analysis was performed on the online tool of UALCAN based on sample types. UALCAN (http://ualcan.path.uab.edu, accessed on 10 February 2022) is a user-friendly, interactive web portal for analyzing TCGA gene expression data in depth [49].

### 4.5. Analysis of Gene Expression and Mutation Alterations Using cBioPortal 

The cBioPortal for Cancer Genomics (http://www.cbioportal.org, accessed on 10 February 2022) is a well-known and frequently used online website that allows large-scale cancer genomics datasets to be visualized, analyzed, and downloaded [50]. In our study, cBioPortal was used to analyze the frequency of mutations and copy number alterations of *BRCA1*/*2* with appropriate parameter settings.

### 4.6. Correlation Analysis of BRCA1/2 and Breast Cancer Stage

The ggplot2 program was used to perform a correlation analysis of *BRCA1/2* expression and cancer stage. The ROC curve for diagnosis analysis was created using R packages. This section’s clinical data were obtained from the TCGA database.

### 4.7. Prognosis Analysis Using PrognoScan 

PrognoScan (http://dna00.bio.kyutech.ac.jp/PrognoScan/, accessed on 10 February 2022) is a widely used online database for meta-analysis of the prognostic significance of genes [51]. In this study, PrognoScan was used to analyze the association of *BRCA1*/*2* expression with the survival in breast cancer patients.

### 4.8. Survival Analysis Using Kaplan–Meier Plotter 

The Kaplan–Meier plotter (https://kmplot.com/analysis/, accessed on 10 February 2022) is an online application tool that can evaluate the impact of the expression of more than 54,000 genes on the survival of 21 different cancer types [52]. It was used to analyze the relationship between *BRCA1/2* expression and the survival of patients with breast cancer in this study.

### 4.9. Survival and Correlation Analysis Using R2 

R2 (https://hgserver1.amc.nl/cgi-bin/r2/main.cgi, accessed on 10 February 2022) is an online tool that can be used for a variety of genomics research and visualization tasks. In this study, we used R2 platform to analyze the correlations between *BRCA1*/*2* expression and the survival of patients with breast cancer.

### 4.10. GO and KEGG Enrichment Analysis 

To investigate probable biological activities and signaling pathways influenced by *BRCA1/2*, we used the package R “cluster profile” to perform Gene Ontology (GO) and Kyoto Encyclopedia of Genes and Genomes (KEGG) analyses on co-expressed genes. GO analysis is a commonly used method for large-scale functional enrichment research; gene functions can be classified into biological process (BP), cellular component (CC), and molecular function (MF).

### 4.11. Analysis of BRCA1/2 Interacting Genes and Proteins 

The GeneMANIA database (http://www.genemania.org, accessed on 10 February 2022) was applied to construct the gene–gene interaction network of *BRCA1/2*. The STRINGdb analysis tool (https://string-db.org/cgi/input.pl, accessed on 10 February 2022) is a database of known and predicted protein interactions and was used to construct a protein–protein interaction (PPI) network of *BRCA1/2*.

### 4.12. Statistical Analysis

R (v.3.6.3) was used to perform the statistical analysis. Differences between groups were compared using the Wilcoxon rank-sum test or Student’s *t* test, as appropriate. Pearson or Spearman correlation tests, as appropriate, were used to determine correlations. Survival curves were generated using PrognoScan, the Kaplan–Meier Plotter, and the R2. All results were shown as *p* values of the log-rank test. They were created, and log-rank tests were performed to identify the significance of the difference between the survival curves, and differences with a *p* < 0.05 were considered statistically significant.

## Figures and Tables

**Figure 1 ijms-23-03754-f001:**
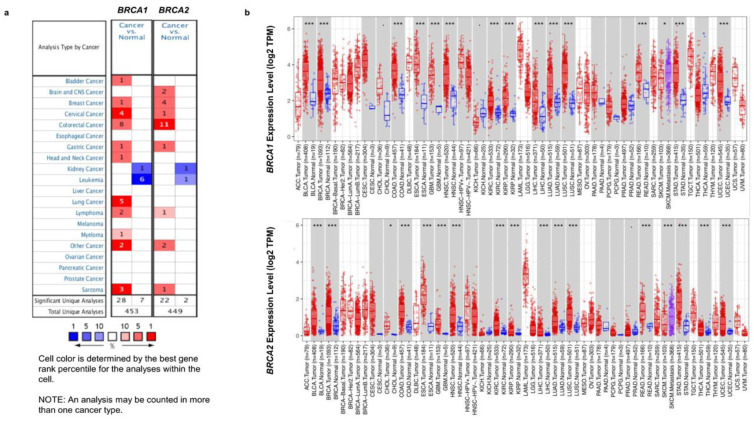
Expression of *BRCA1/2* mRNA in cancer vs. normal (Oncomine and TCGA database). (**a**) This graphic generated by Oncomine (https://www.oncomine.org/resource/login.html, accessed on 10 February 2022) indicates the numbers of datasets with statistically significant (*p* < 0.01) mRNA overexpression (red) or downexpression (blue) of *BRCA1/2* (different types of cancer vs. corresponding normal tissue). The threshold was designed with the following parameters: *p* value of 1 × 10^−4^, fold change of 2, and gene ranking of 10%. (**b**) Human *BRCA1/2* expression levels in different tumor types from TCGA database were determined by TIMER 2.0 (http://timer.cistrome.org, accessed on 10 February 2022) (* *p* < 0.05; *** *p* < 0.001).

**Figure 2 ijms-23-03754-f002:**
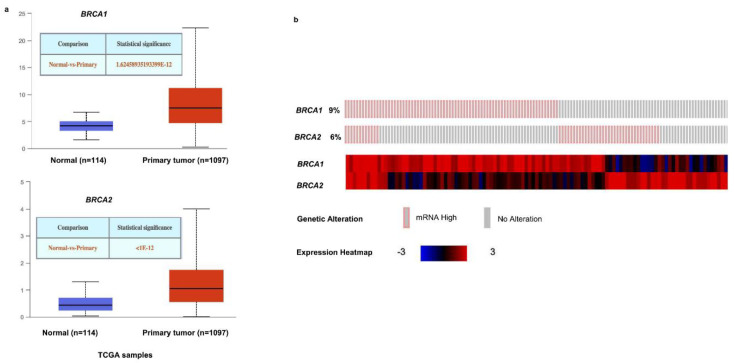
Expression of *BRCA1/2* in BRCA patients. (**a**) Expression of *BRCA1/2* in BRCA based on sample types by UALCAN (http://ualcan.path.uab.edu/index.html, accessed on 10 February 2022). (**b**) Expression of *BRCA1/2* in BRCA by cBioPortal database (https://www.cbioportal.org, accessed on 10 February 2022) (breast invasive carcinoma (TCGA, Cell 2015)).

**Figure 3 ijms-23-03754-f003:**
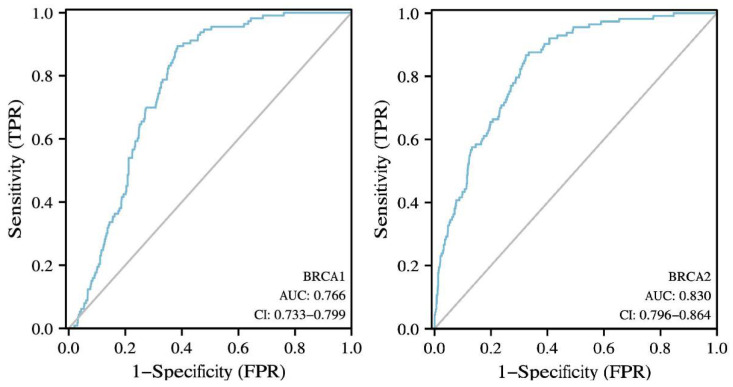
The diagnostic value of *BRCA1/2* in BRCA patients. The diagnostic value of *BRCA1/2* in BRCA patients. AUC: area under curve.

**Figure 4 ijms-23-03754-f004:**
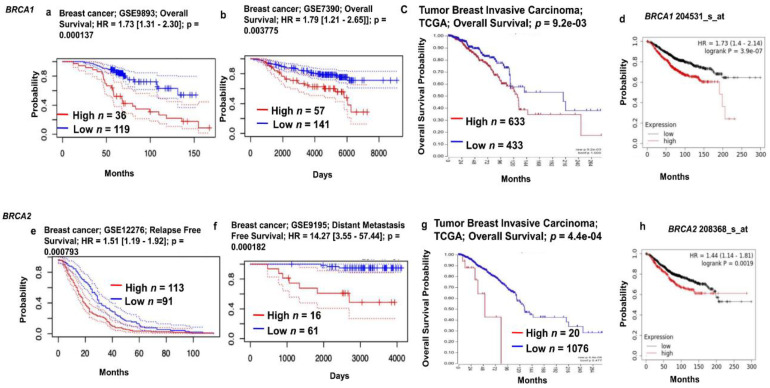
Relationship between *BRCA1/2* mRNA expression and clinical outcomes in breast cancer patients (PrognoScan Database, R2: Kaplan Meier Scanner and Kaplan–Meier Plotter). (**a**,**b**,**e**,**f**) The survival curve comparing the patient with high (red), and low (blue) expression of *BRCA1/2* was plotted from the PrognoScan database (http://dna00.bio.kyutech.ac.jp/PrognoScan/, accessed on 10 February 2022) in breast cancer patients. The threshold of cox *p* value < 0.05. (**c**,**g**) The survival curve comparing the patient with high (red) and low (blue) expression of *BRCA1/2* was plotted from R2: Kaplan Meier Scanner (https://hgserver1.amc.nl/cgi-bin/r2/main.cgi, accessed on 10 February 2022) in TCGA breast cancer patients. The threshold of cox *p* value < 0.05. (**d**,**h**) The survival curve comparing the patient with high (red) and low (black) expression of *BRCA1/2* was plotted from Kaplan–Meier Plotter (https://kmplot.com/analysis/, accessed on 10 February 2022) in breast cancer patients. The threshold of cox *p* value < 0.05.

**Figure 5 ijms-23-03754-f005:**
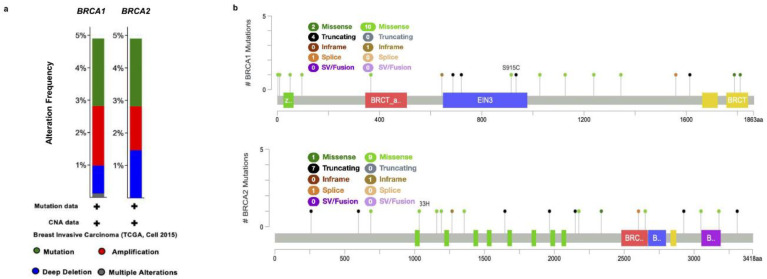
*BRCA1/2* alteration frequency of mutations and copy number alterations (CNAs) in Breast Cancer (cBioPortal web). (**a**) The alteration frequency of the *BRCA1/2* signature was determined cBioPortal (http://www.cbioportal.org, accessed on 10 February 2022). The alteration frequency included mutations (green), amplifications (red), deep deletions (blue), and multiple alterations (grey). (**b**) Summary of mutations in *BRCA1/2* in BRCA from TCGA, cell 2015.

**Figure 6 ijms-23-03754-f006:**
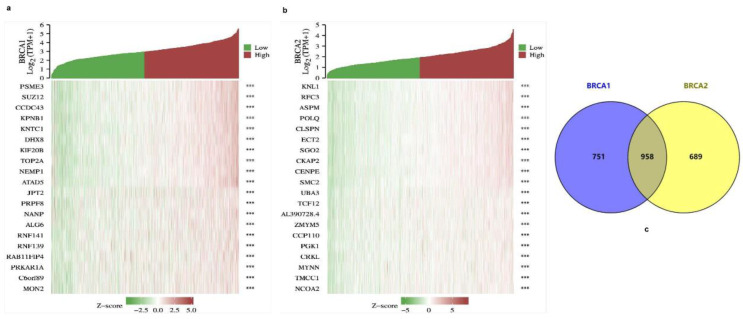
Co-expressed genes of *BRCA1/2*. (**a**,**b**) The top 10 genes with positive and negative co-expression of *BRCA1/2* in the TCGA database according to heat map and Venn map. (**c**) Intersection co-expression genes of *BRCA1/2*. Note: |r| > 0.4 and *p* < 0.001.

**Figure 7 ijms-23-03754-f007:**
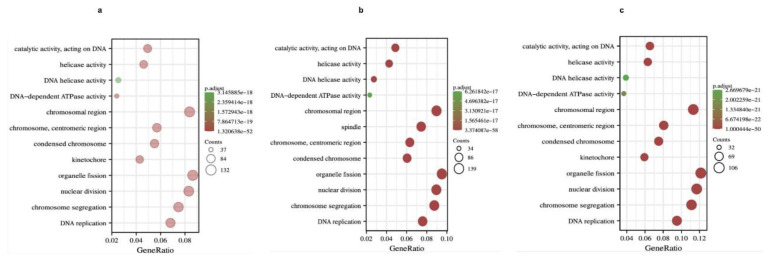
GO analysis of *BRCA1*, *BRCA2* co-expression genes, and intersection co-expression genes among *BRCA1/2*. (**a**) GO analysis of *BRCA1*, (**b**) GO analysis of *BRCA2*, (**c**) GO analysis of co-expression genes, and intersection co-expression genes among *BRCA1/2*. GO: Gene Ontology.

**Figure 8 ijms-23-03754-f008:**
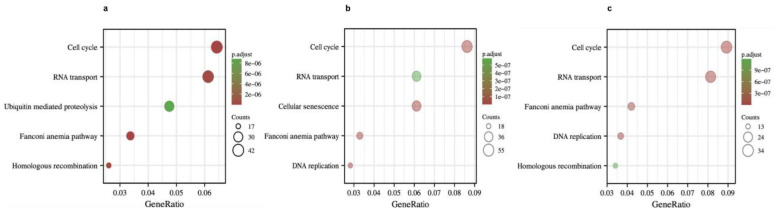
KEGG pathway enrichment analysis of *BRCA1*, *BRCA2* co-expression genes, and intersection co-expression genes among *BRCA1/2*. KEGG pathway enrichment analysis of *BRCA1* (**a**), *BRCA2* (**b**) co-expression genes, and intersection co-expression genes among *BRCA1/2* (**c**). KEGG: Kyoto Encyclopedia of Genes and Genome.

**Figure 9 ijms-23-03754-f009:**
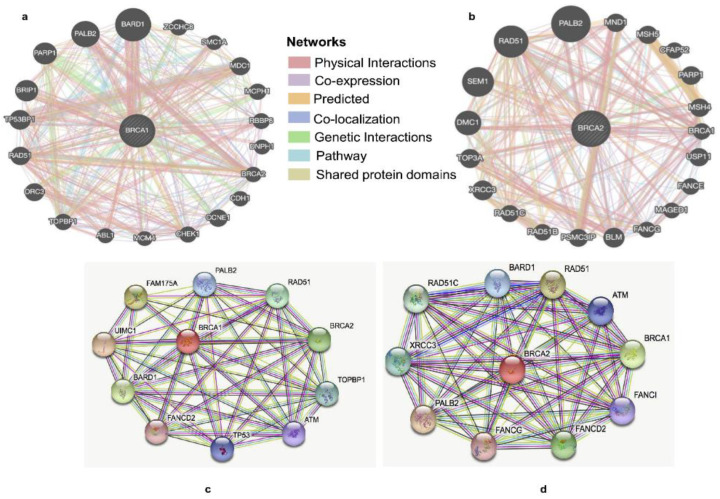
Identification of *BRCA1/2* interacting genes and proteins. (**a,b**) The gene–gene interaction network of *BRCA1/2* was constructed using GeneMania. (**c,d**) The protein–protein interaction network of *BRCA1/2* was generated using STRING db.

**Table 1 ijms-23-03754-t001:** The expression of *BRCA1/2* in BRCA in the Oncomine database.

Gene Name	Types of BRCA vs. Normal	Fold Change	*t* Test	*p* Value	Reference
*BRCA1*	Intraductal Cribriform Breast Adenocarcinoma vs. Normal	3.454	15.862	1.92 × 10^−6^	TCGA Breast
*BRCA2*	Mucinous Breast Carcinoma vs. Normal	3.750	9.344	3.22 × 10^−5^	TCGA Breast
	Invasive Breast Carcinoma vs. Normal	2.602	9.978	5.91 × 10^−18^	TCGA Breast
	Invasive Ductal Breast Carcinoma vs. Normal	3.214	14.586	1.46 × 10^−25^	TCGA Breast
	Invasive Lobular Breast Carcinoma vs. Normal	2.124	6.314	9.36 × 10^−9^	TCGA Breast

TCGA: The Cancer Genome Atlas.

**Table 2 ijms-23-03754-t002:** The relationships between *BRCA1/2* expression and clinicopathological features in Breast Cancer.

Variables	No.	BRCA1 (*p* Value)	BRCA2 (*p* Value)
**Age**		NS	*p* < 0.01
≤60	601		
>60	482		
**T-Stage**		NS	NS
T1	277		
T2	629		
T3	139		
T4	35		
**N-Stage**		*p* < 0.05	*p* < 0.01
N0	514		
N1	358		
N2	116		
N3	76		
**M-Stage**		NS	NS
M0	902		
M1	20		
**ER status**		*p* < 0.01	*p* < 0.001
−	240		
+	793		
**PR status**		*p* < 0.001	*p* < 0.001
−	342		
+	688		
**HER2 status**		NS	*p* < 0.01
−	558		
+	157		
**PAM (50)**		*p* < 0.001	*p* < 0.001
Lum A	562		
Lum B	204		
Her 2	82		
Basal	195		
Normal	40		

T stage: primary tumor, N stage: regional lymph nodes, M stage: distant metastasis, *ER* status: estrogen receptor, *PR* status: progesterone receptor, *HER2* status: human epidermal growth factor receptor 2, PAM (50): prediction analysis of microarray 50, Lum A: luminal A, Lum B: luminal B, Her2: Her2-enriched, Basal: basal-like, Normal: normal-like, NS: no significance.

## Data Availability

The datasets created and/or analyzed during this investigation are available to the public. In the Materials and Methods section, each portal’s persistent web link has been supplied.

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
