# Peer review of "BRCA1/2 Serves as a Biomarker for Poor Prognosis in Breast Carcinoma"

_ijms, 2022, doi:10.3390/ijms23073754_

Round 1

Reviewer 1 Report

In this report, the authors correlated expression levels of BRCA1/2 which clinical outcomes. The study provides some insight on the roles of these two breast cancer susceptibility genes in tumor development and progression. However, the authors need to provide more description and interpretation of their studies. At times, their conclusions also seem contradictory. Please see my comments below.

Major comments

  1. Line 305: “In this study, we found increased BRCA1/2 mRNA levels were associated with overall survival (OS) and that over expression of BRCA1/2 was significantly related to worse overall survival (P< 0.001)”. This sentence from the discussion summarizes two apparent contradictory conclusions. How can you conclude that high mRNA levels are associated with better survival while over-expression is associated with poor survival. Isn’t expression level deduced from mRNA levels? Or are you looking at protein levels? This also has to be explained better in Results sections 3.1 and 3.2. These two sections appear to contradict each other.
  2. One question would be whether the over-expressed BRCA1/2 is WT or mutant. The authors appear to check this in Results section 3.4. But why are the authors only looking at breast invasive carcinoma mutations? This does not appear to correlate well with the present study. Most TCGA studies give both expression levels and genome alteration (e.g. mutation, deletion, CNV, etc). The authors should download the mutation data for the same studies for which they report expression data and check whether over-expression correlates with increased mutation or whether WT BRCA1/2 is over-expressed.
  3. Line 290. “Several studies have reported that BRCA1 and BRCA2 carriers have varied clinical characteristics. Herein, our study found that over expression of BRCA1 was related to lymph node stage, ER status, PR status and PAM 50 subtypes (P<0.05), and over expres- 292 sion of BRCA2 was related to age, lymph node stage, ER stage, PR status, HER-2 status, 293 and PAM 50 subtypes (P><0.01). ><0.05) and over expression of BRCA2 was related to age, lymph node stage, ER stage, PR status, HER-2 status, 293 and PAM 50 subtypes (P<0.01). ><0.01)” First, please reference the “several studies” you refer to. Second, here you are suggesting that BRCA1 over-expression levels is correlated with different outcomes than BRCA2 over-expression levels while earlier you combined both genes together (see point 1). This is confusing. You need to parse out the outcomes of BRCA1 and BRCA2 independently. I suggest you make a figure in the discussion with independent correlations between BRCA1 and BRCA2 expression levels and altered processes. Then determine whether there are any common processes related to over-expression of both genes. That’s why it is important that in the introduction you spell out in more detail the different functions of BRCA1 and BRCA2 which are not identical (see minor point 1 below).

Minor comments

  1. In the introduction, please expand on BRCA1, BRCA2, and PALB2 function with some references. For example, BRCA2 is essential for loading RAD51 while BRCA1 seems to have a more accessory function. Additionally, PALB2 bridges the interaction between BRCA2 and BRCA1 and facilitates their RAD51 loading activity.
  2. Line 214. What is meant by deep deletions? You mean homozygous deletion?

Author Response

Dear reviewer,

Authors first heart-fully thanking the  reviewer for their insightful comments on our manuscript titled “BRCA1/2 serves as a biomarker for poor prognosis in Breast Carcinoma”. These comments are all valuable and very helpful for revising and improving our paper, as well as the important guiding significance to our research. We have carefully reviewed each comment provided by the reviewer and have revised the manuscript accordingly. Our responses are given below in a point-by-point manner. Changes to the manuscript are shown in yellow/. Kindly check the attached file. Thank you.

Sincerely,

Kyoung Sik Park

Reviewer 2 Report

Title: BRCA1/2 serves as a biomarker for poor prognosis in Breast 2 Carcinoma

Abstract: The authors employ many bioinformatics methods to perform a systematic multiomodal analysis to expound BRCA1/2 functions as prognostic biomarkers in breast cancer. The authors used platforms like On-13 comine, TIMER 2.0, UALCAN, and cBioportal to understand the expression of BRCA1/2 genes. Further the authors determined 15 the relationship between BRCA1/2 mRNA expression and prognosis in cancer patients. The authors analysed various pathays like GO and KEGG  USINGa pathway enrichment analysis to elucidate the potential biological functions of the co-expression genes of BRCA1/2. The authors found that The BRCA1/2 mRNA levels in breast cancer tissues 22 was considerably higher than in normal tissues. The authors concluded that their study provides suggestive evidence of the prognostic role of BRCA1/2 in breast cancer and the therapeutic target for breast cancer.

Major comments.

The review article is a highly interesting topic. The findings are amazing and are presented by the authors in an impeccable way. The authors have done an excellent job in presenting the research findings and explaining them as well. The pictorial representations of the data is excellent as well. According to the reviewer, the manuscript can be accepted with some improvements. The reviewer would like to add some major and minor corrections:

Major corrections:

  1. Add a section where you can summarise some interesting pre-clinical studies on using BRCA1/2 gene as a target for breast cancer in vivo model. Example: https://www.tandfonline.com/doi/abs/10.1080/13543784.2020.1818067?journalCode=ieid20 

 Minor corrections:

  1. Line 272, there should be a space between 12 fold
  2. Line 275- "Previous studies evaluated 275 the risks of BRCA1/2 positive families, respectively" Please clarify
  3. Line 276-279 Please clarify the meaning. 
  4. Line 290-293 Please clearly elucidate the research findings. Please explain and elaborate. 

Author Response

(The authors gave the same response as above.)

Round 2

Reviewer 1 Report

The authors have made the requested changes. This reviewer is satisfied.